# Efficient Diffusion Models via Time Step Optimization with Consistent Training and Inference Constraints

**Binrui Wu** [* 2 1]   **Zihao Cheng** [* 1]   **Yuesen Liao** [* 1 4]   **Weizhong Zhang** [1 3]

## Abstract

Diffusion probabilistic models(DPMs)' sampling process is often inefficient, requiring hundreds to thousands of iterative steps to accurately approximate the diffusion trajectory. This inefficiency limits their practical applicability. Although recent advances in sampling efficiency—such as numerical solvers for diffusion ordinary differential equations (ODEs) have made progress, significant challenges remain: training-free numerical solvers suffer from the suboptimality of manually designed timestep selection rules and the inherent inconsistency between the forward diffusion process (typically involving thousands of steps) and the reverse denoising process (usually limited to tens of steps). Since timestep selection is inherently a discrete problem and cannot be optimized via gradients, we propose an innovative approach—reparameterizing the timestep scheduling through probabilistic masking, thereby enabling gradient-based optimization of sampling timesteps. To circumvent backpropagation, we employ policy gradient methods. Furthermore, to address the inconsistency between forward diffusion and reverse denoising, we extend this framework into a bilevel optimization paradigm: the inner loop performs lightweight training on the model at specific timesteps determined by the outer mask to align forward and reverse processes, while the outer loop optimizes the timestep distribution via probabilistic masking and policy gradient based on generation quality. Under mild assumptions, we theoretically analyze the convergence of the proposed algorithm. Extensive experiments demonstrate that this framework effectively enhances sampling efficiency and generation quality while maintaining compatibility with various DPM architectures.

## 1. Introduction

Diffusion Probabilistic Models (DPMs) (Sohl-Dickstein et al., 2015; Song et al., 2021b; Ho et al., 2020; Song & Ermon, 2019) have revolutionized the landscape of visual content generation, spanning diverse domains such as image synthesis (Dhariwal & Nichol, 2021; Rombach et al., 2022; Saharia et al., 2022; Ramesh et al., 2022), video production (Blattmann et al., 2023; Ho et al., 2022a; Gupta et al., 2023; Ho et al., 2022b; Luo et al., 2023), and 3D modeling (Poole et al., 2022; Tang et al., 2023; Jensen et al., 2014). These models operate through an iterative process that systematically transforms a data distribution into a noise distribution via a predefined stochastic differential equation (SDE), with generative capabilities achieved through learning to reverse the noise-adding process. Despite their impressive performance, DPMs typically necessitate hundreds to thousands of denoising steps to accurately approximate the diffusion trajectory, resulting in substantially slower sampling speeds compared to single-step generative approaches like GANs (Goodfellow et al., 2014) and VAEs (Kingma & Welling, 2013). This inherent inefficiency poses a significant challenge for the widespread adoption and scalability of DPMs in practical applications.

In recent years, numerous efficient sampling techniques have been proposed for DPMs, among which two categories stand out due to the probability flow ODE inherent in the reverse process of DPMs. The first category is diffusion model distillation (Meng et al., 2023; Sauer et al., 2023; Salimans & Ho, 2022), which achieves "step-skipping" by aligning a student trajectory with fewer steps to a more precise teacher trajectory. However, these methods require expensive retraining or fine-tuning of the entire network and face difficulties with conditional sampling.

The second category of methods involves designing more accurate numerical solvers for diffusion ODEs (Lu et al., 2022a; Song et al., 2021a; Lu et al., 2022b). These methods typically do not require retraining the neural network but often underperform in terms of generation quality, especially

---
[*]Equal contribution  [1]Fudan University [2]Alibaba International Digital Commerce Group [3]Shanghai Key Laboratory of Intelligent Information Processing [4]Xiaohongshu Inc.. Correspondence to: Weizhong Zhang <weizhongzhang@fudan.edu.cn>.

*Proceedings of the $43^{rd}$ International Conference on Machine Learning*, Seoul, South Korea. PMLR 306, 2026. Copyright 2026 by the author(s).

under a limited number of steps. We attribute this suboptimal performance primarily to two reasons. One is these methods (Karras et al., 2022; Lu et al., 2022a; Song et al., 2021a) always first discretize the time space into several interval using heuristic strategyies and then primarily focus on developing or leveraging advanced numerical ODE solvers to better solve probability flow ODEs in each interval.

We argue that time space discretization is important and such heuristic strategies often lead to suboptimal generation quality, and it is challenging to find a universal rule that can adapt to different solvers and models. The other is the inconsistency between the training and inference time schedules. That is these approaches rely on a pre-trained denoising model, which is trained on a complete forward process comprising thousands or even continuous timesteps (taking DDPM (Ho et al., 2020) as an example, which discretizes the diffusion process into 1000 timesteps with uniform training during the training phase). This process enforces the model to fit the noise/score function at all timesteps. However, during inference, only a few timesteps are utilized. This discrepancy implies that the effort invested in fitting the noise at numerous unused timesteps is largely wasted and these efforts should have been used on the selected time steps to improve the performance.

To address the first issue, we first construct a candidate set of timesteps, which is indexed by a binary mask list (where a mask value of 1 indicates that the timestep is used during sampling, while 0 means it is skipped). We then reparameterize mask as a Bernoulli distribution, i.e., $\boldsymbol{m} \sim \mathrm{Bern}(\boldsymbol{s})$, where $\boldsymbol{s}$ is the probability vector, thereby transforming the originally discrete, non-differentiable problem into a continuous and differentiable optimization task. We define a loss function based on the distance between the generated distribution under the time schedule defined by the random mask and the real data distribution to evaluate the quality of the learned time schedule. Then, for optimizing the time schedule, we develop a policy gradient estimator to avoid expensive backpropagation.

Furthermore, to mitigate the training-inference misalignment, we extend our above optimization method into a bilevel optimization framework that simultaneously selects optimal timesteps during sampling while aligning the forward diffusion and reverse denoising processes through auxiliary training in the inner loop. Specifically, the inner loop finetune the model under the timestep schedule given by the outer loop to minimize the gap between training and inference; the outer loop continues to parameterize the timestep schedule as a list of Bernoulli distributions, evaluates the generative quality of the model finetuned in the inner loop and optimizes the time schedule using policy gradient estimator. We extensively validate our framework across multiple datasets(e.g., CIFAR10 (Krizhevsky et al., 2009),

ImageNet (Russakovsky et al., 2015)) and samplers(e.g., DDIM (Song et al., 2021a), iPNDM (Liu et al., 2022)), demonstrating its effectiveness. Our key contributions are summarized as follows:

- We introduce an optimization framework employing probabilistic timestep masking, which adaptively learns the optimal sampling schedule in diffusion models via policy gradient optimization.

- We identify the training-inference discrepancy inherent in diffusion models and accordingly extend our optimization framework into a bilevel learning paradigm, which fully unleashes the generative potential of diffusion models under few-step sampling scenarios.

- We provide a theoretical convergence analysis of the proposed algorithm and establish rigorous mathematical guarantees for its convergence.

- We conduct extensive experiments on multiple datasets and samplers, achieving promising results that validate the effectiveness of our approach.

## 2. Background

Diffusion Probabilistic Models (DPMs) (Sohl-Dickstein et al., 2015; Ho et al., 2020; Song et al., 2021b) define a forward diffusion process that gradually corrupts a clean data sample $\mathbf{x}_0$ drawn from an unknown distribution $q$ into a sequence of noisy variables $\{\mathbf{x}_t\}_{t \in [0,T]}$, where $T > 0$ denotes the total diffusion time. At each intermediate step $t$, the conditional distribution of $\mathbf{x}_t$ given $\mathbf{x}_0$ follows a Gaussian transition:

$$q(\mathbf{x}_t|\mathbf{x}_0) = \mathcal{N}(\alpha_t \mathbf{x}_0, \sigma_t^2 \mathbf{I}), \tag{1}$$

where $\alpha_t$ and $\sigma_t$ are differentiable functions of time $t$. The pair $(\alpha_t, \sigma_t)$ constitutes the noise schedule of DPMs and is designed such that the signal-to-noise ratio (SNR) $\alpha_t^2/\sigma_t^2$ decreases monotonically (Kingma et al., 2021). By the final timestep $T$, the distribution converges to an isotropic Gaussian, i.e., $\mathbf{x}_T \sim \mathcal{N}(0, \sigma_T^2 \mathbf{I})$.

To approximate the data distribution $q(\mathbf{x}_0)$, DPMs learn a reverse denoising process that recovers the information lost in the forward diffusion. A neural network $\varepsilon_\theta(\mathbf{x}_t, t)$ is trained to predict the injected noise by minimizing

$$\mathbb{E}_{\mathbf{x}_0, \varepsilon, t}\left[\omega(t)\|\varepsilon_\theta(\mathbf{x}_t, t) - \varepsilon\|_2^2\right], \tag{2}$$

where $\mathbf{x}_t = \alpha_t \mathbf{x}_0 + \sigma_t \varepsilon$ with $\varepsilon \sim \mathcal{N}(0, \mathbf{I})$, $t \sim \mathcal{U}[0, T]$. $\omega(t) > 0$ serves as a time-dependent weighting term.

Sampling from a trained DPM can be interpreted as solving the corresponding diffusion ordinary differential equation

(ODE) (Song et al., 2021b;a):

$$\frac{d\mathbf{x}_t}{dt} = f(t)\mathbf{x}_t - \frac{1}{2}g^2(t)\nabla_{\mathbf{x}}\log q_t(\mathbf{x}_t), \quad \mathbf{x}_T \sim q_T(\mathbf{x}_T),$$
(3)

where $f(t)$ and $g(t)$ denote the drift and diffusion coefficients defined as

$$f(t) = \frac{d\log\alpha_t}{dt}, \quad g^2(t) = \frac{d\sigma_t^2}{dt} - 2\frac{d\log\alpha_t}{dt}\sigma_t^2.$$
(4)

Because $\varepsilon_\theta(\mathbf{x}_t, t)$ approximates the scaled score function $-\sigma_t\nabla_{\mathbf{x}}\log q_t(\mathbf{x}_t)$, (Song et al., 2021b) formulated the following parameterized diffusion ODE:

$$\frac{d\mathbf{x}_t}{dt} = f(t)\mathbf{x}_t + \frac{g^2(t)}{2\sigma_t}\varepsilon_\theta(\mathbf{x}_t, t), \quad \mathbf{x}_T \sim q_T(\mathbf{x}_T).$$
(5)

This can be equivalently rewritten as

$$d\left(\frac{\mathbf{x}_t}{\alpha_t}\right) = \varepsilon_\theta(\mathbf{x}_t, t)\, d\left(\frac{\alpha_t}{\sigma_t}\right).$$
(6)

In discrete numerical solvers such as DDIM (Song et al., 2021a), the continuous trajectory is discretized into a timestep schedule $\{t_N, t_{N-1}, \ldots, t_0\}$, where $t_N = T$ and $t_0 = 0$. Each update step follows the scheme:

$$\frac{\mathbf{x}_{t_{i-1}}}{\alpha_{t_{i-1}}} = \frac{\mathbf{x}_{t_i}}{\alpha_{t_i}} + \left(\frac{\sigma_{t_{i-1}}}{\alpha_{t_{i-1}}} - \frac{\sigma_{t_i}}{\alpha_{t_i}}\right)\varepsilon_\theta(\mathbf{x}_{t_i}, t_i),$$
(7)

with i = 1, ..., N-1.

## 3. Related Work

**Diffusion Probabilistic Models**. Diffusion Probabilistic Models (DPMs) (Ho et al., 2020; Sohl-Dickstein et al., 2015; Song & Ermon, 2019; Song et al., 2021b) progressively transform real data into noise through Gaussian perturbations and generate samples from noise via sequential denoising steps. This iterative refinement process offers a stable and expressive generative framework, enabling models to capture complex data distributions more effectively than earlier likelihood- or adversarial-based methods. Recent studies (Dhariwal & Nichol, 2021; Rombach et al., 2022; Saharia et al., 2022; Ramesh et al., 2022) have demonstrated that DPMs surpass traditional generative models such as GANs (Goodfellow et al., 2014) and VAEs (Kingma & Welling, 2013) in image synthesis quality, producing high-fidelity and diverse samples with improved training stability. These successes have established diffusion models as a dominant paradigm for modern generative modeling.

**Efficient Sampling for DPMs**.

Compared to GANs (Goodfellow et al., 2014), a significant drawback of diffusion models lies in their inefficient sampling. Recent studies (Sauer et al., 2023; Song et al.,

2021a; Liu et al., 2022; Meng et al., 2023) have focused on improving the sampling efficiency of diffusion models, among which knowledge distillation (Sauer et al., 2023; Salimans & Ho, 2022; Meng et al., 2023) has emerged as an effective approach. The core idea is to train a student model to approximate the multi-step inference output of a teacher model with fewer steps. This paradigm leverages the observation that the reverse diffusion trajectory is highly redundant, allowing a compact student model to retain most of the generative capacity while discarding unnecessary intermediate states. For instance, ADD (Sauer et al., 2023) enhances the distillation process by introducing adversarial loss, while PD (Salimans & Ho, 2022) employs a progressive distillation strategy to gradually improve sampling efficiency. However, these methods typically require substantial additional training overhead, since the student must repeatedly mimic multi-step trajectories produced by the teacher model.

Another popular approach involves discretizing the reverse-time probability flow ordinary differential equations (ODEs) (Lu et al., 2022a; Sabour et al., 2024), which can be divided into two orthogonal directions. The first focuses on designing more advanced ODE solvers. For example, DPM-Solver (Lu et al., 2022a) analytically computes the linear part of the ODE, transforming the solution into an exponentially weighted integral of the neural network, whereas iPNDM (Liu et al., 2022) proposes pseudo-numerical methods through analysis of differential equations on manifolds. These solvers attempt to better approximate the reverse process with large step sizes while preserving stability, and reduce the number of function evaluations required during inference. Nevertheless, these methods often rely on hand-crafted sampling schedules (Karras et al., 2022), which may not be optimal and can lead to suboptimal trade-offs between accuracy and efficiency across different datasets or noise regimes.

This leads to another research direction: optimizing the sampling time step scheduling (Xue et al., 2024; Xia et al., 2024; Tong et al., 2024). Recent works such as DMN (Xue et al., 2024) derive the discrete-time schedule by minimizing the upper bound of global truncation error, while AYS (Sabour et al., 2024) leverages stochastic calculus to identify dataset-aware optimal schedules. Furthermore, (Xia et al., 2024) introduces a reinforcement learning-based scheduler predictor to dynamically determine the optimal denoising steps. These scheduling approaches highlight that, even with strong solvers, the choice of time discretization remains a critical factor influencing the final efficiency–quality balance of diffusion model sampling.

Despite these advances, solver design and time-step allocation are often treated separately, leaving their interaction with denoising dynamics underexplored.

# 4. Methods

In this section, we present the details of our proposed algorithm. First, in Section 4.1, we introduce an efficient algorithm to optimal sampling schedule. Next, Section 4.2 provides a detailed analysis of the training-inference inconsistency phenomenon inherent in diffusion model sampling. Finally, we extend the algorithm proposed in Section 4.1 into the bielievl framework to mitigate this inconsistency in Section 4.3.

## 4.1. Timestep Selection through Probabilistic Masking

**Problem Formulation.** Given a denoising network $\epsilon_\theta(x_t, t)$ trained by optimizing the objective Eq. (2), where $t$ is sampled from a certain distribution on thousands of discrete timesteps or on the continuous time range $[0, T]$. Traditional efficient sampling methods would always follow a handcrafted time schedule, denoted as

$$\rho = \{t_{i_n}, t_{i_{n-1}}, \ldots, t_{i_0}\},$$

where $N$ is the total steps and $N > i_n > i_{n-1} > \cdots > i_0 > 0$, then solve the following inverse probability flow ODE along this schedule:

$$\frac{\mathrm{d}x_t}{\mathrm{d}t} = \frac{\mathrm{d}\log\alpha_t}{\mathrm{d}t}x_t + \frac{1}{2\sigma_t}\left(\frac{\mathrm{d}\sigma_t^2}{\mathrm{d}t} - 2\frac{\mathrm{d}\log\alpha_t}{\mathrm{d}t}\sigma_t^2\right)\epsilon_\theta(x_t, t). \tag{8}$$

We contend that this kind of approach is always suboptimal for the following two considerations:

- Handcrafted schedule in the sampling process doesn't guarantee best performance;

- Different samplers may not share the same sampling time steps, which is supported by our empirical results given in Tab2:while the EDM (Karras et al., 2022) rule achieves superior generation quality when applied to the iPNDM (Liu et al., 2022) sampler, the LogSNR (Lu et al., 2022a) rule demonstrates significantly better performance on Uni-PC (Zhao et al., 2023) under identical evaluation conditions.

The above analysis demonstrates that existing handcrafted rules struggle to accommodate diverse sampling strategies effectively. Our target is to develop a method for automatically selecting the optimal timesteps, which is a high-dimensional combinatorial optimization problem that becomes very complex when the amount of candidate timesteps is large. If we aim to find a $K$ steps sampling from $N$ original steps, then the complexity is $C_{N-1}^K$(the first and last steps are fixed). To tackle this problem, we assign a mask $\boldsymbol{m} = \{m_1, m_2, \ldots, m_{N-1}\}$ on timesteps $\{t_1, t_2, \ldots, t_{N-1}\}$ which is parameterized by $\boldsymbol{s} = \{s_1, s_2, \ldots, s_{N-1}\}$, where $m_i \sim \text{Bern}(s_i)$, and

$||\boldsymbol{s}||_1 \leq K$. This is reminiscent of mask-based optimization in model pruning (Li et al., 2026). By training the parameter $\boldsymbol{s}$ to convergence, we derive the optimal time schedule. The training target can be formulated as:

$$\min_{\boldsymbol{s}} \ \Phi(\boldsymbol{s}) = \mathbb{E}_{\boldsymbol{m}\sim p(\boldsymbol{m}|\boldsymbol{s})}\mathcal{L}_{\text{eval}}(\boldsymbol{\theta}^*, \boldsymbol{m}),$$
$$\text{s.t. } \boldsymbol{s} \in \mathcal{C} \triangleq \{\boldsymbol{s} \in [0, 1]^{N-1} \mid ||\boldsymbol{s}||_1 \leq K\}. \tag{9}$$

where $\boldsymbol{\theta}^*$ is the pretrained denoising model, and $\mathcal{L}_{eval}(\boldsymbol{\theta}^*, \boldsymbol{m})$ evaluates the sample quality of the model using the time schedule corresponding to mask $\boldsymbol{m}$. This optimization paradigm is named as PCS(probabilistic coreset selection). In this paper, we define $\mathcal{L}_{eval}(\boldsymbol{\theta}^*, \boldsymbol{m})$ to be

$$\mathcal{L}_{\text{eval}}(\boldsymbol{\theta}^*, \boldsymbol{m}) =$$
$$\mathbb{E}_{x_{t_N}\sim\mathcal{N}(0,I)}\mathcal{L}_{\text{dist}}\Big(F_{\boldsymbol{\theta}^*}(x_{t_N}; \boldsymbol{m}; \phi), F_{\boldsymbol{\theta}^*}(x_{t_N}; \mathbf{1}; \psi)\Big), \tag{10}$$

where $\mathcal{L}_{dist}$ can be general metric to evaluate the distance between two distributions. $F_{\boldsymbol{\theta}^*}(x_{t_N}; \boldsymbol{m}; \phi)$ and $F_{\boldsymbol{\theta}^*}(x_{t_N}; \mathbf{1}; \psi)$ generate samples along under the masked time schedule $\boldsymbol{m}$ and the full schedule, respectively. We allow different samplers $\phi$ and $\psi$ to be used in the masked and full time schedules. For example, if we let $\phi$ be the DDIM sampler, i.e., Eq. (7), given a sampled mask $\boldsymbol{m}$, $F_{\boldsymbol{\theta}^*}(x_{t_N}; \boldsymbol{m}; \phi)$ would generate samples as follows:

$$\frac{x_{t_{i_{k-1}}}}{\alpha_{t_{i_{k-1}}}} = \frac{x_{t_{i_k}}}{\alpha_{t_{i_k}}} + \left(\frac{\sigma_{t_{i_{k-1}}}}{\alpha_{t_{i_{k-1}}}} - \frac{\sigma_{t_{i_k}}}{\alpha_{t_{i_k}}}\right)\varepsilon_{\boldsymbol{\theta}^*}(x_{t_{i_k}}, t_{i_k}), \tag{11}$$

where k = 1,..., n, $i_n = N$ and $i_0 = 0$ are fixed, $i_{n-1}, i_{n-2}, \ldots, i_1$ are active positions of the mask $\boldsymbol{m}$, i.e., $m_{i_{n-1}} = m_{i_{n-2}} = \cdots = m_{i_1} = 1$.

**Solving the Optimization Problem.** The probabilistic parameterization $\boldsymbol{s}$ of mask $\boldsymbol{m}$ transfers the discrete optimization problem to a continuous one. We adopt the policy gradient estimator to efficiently optimize problem (9). To be precise, the gradient is calculated as:

$$\nabla_{\boldsymbol{s}}\Phi(\boldsymbol{s}) = \nabla_{\boldsymbol{s}}\mathbb{E}_{\boldsymbol{m}\sim p(\boldsymbol{m}|\boldsymbol{s})}\mathcal{L}_{eval}(\boldsymbol{\theta}^*, \boldsymbol{m})$$
$$= \nabla_{\boldsymbol{s}}\sum_{\boldsymbol{m}\sim p(\boldsymbol{m}|\boldsymbol{s})}\mathcal{L}_{eval}(\boldsymbol{\theta}^*, \boldsymbol{m})\,p(\boldsymbol{m}|\boldsymbol{s})$$
$$= \sum_{\boldsymbol{m}\sim p(\boldsymbol{m}|\boldsymbol{s})}\mathcal{L}_{eval}(\boldsymbol{\theta}^*, \boldsymbol{m})\,\frac{\nabla_{\boldsymbol{s}}p(\boldsymbol{m}|\boldsymbol{s})}{p(\boldsymbol{m}|\boldsymbol{s})}p(\boldsymbol{m}|\boldsymbol{s})$$
$$= \sum_{\boldsymbol{m}\sim p(\boldsymbol{m}|\boldsymbol{s})}\mathcal{L}_{eval}(\boldsymbol{\theta}^*, \boldsymbol{m})\,\nabla_{\boldsymbol{s}}\ln p(\boldsymbol{m}|\boldsymbol{s})p(\boldsymbol{m}|\boldsymbol{s})$$
$$= \mathbb{E}_{\boldsymbol{m}\sim p(\boldsymbol{m}|\boldsymbol{s})}\mathcal{L}_{eval}(\boldsymbol{\theta}^*, \boldsymbol{m})\,\nabla_{\boldsymbol{s}}\ln p(\boldsymbol{m}|\boldsymbol{s}). \tag{12}$$

Hence, $\mathcal{L}_{eval}(\boldsymbol{\theta}^*, \boldsymbol{m})\nabla_{\boldsymbol{s}}\ln p(\boldsymbol{m}|\boldsymbol{s})$ is an unbiased estimation of $\nabla_{\boldsymbol{s}}\Phi(\boldsymbol{s})$. Therefore, we can update $\boldsymbol{s}$ by projected stochastic gradient descent:

$$\boldsymbol{s} \leftarrow \mathcal{P}_{\mathcal{C}}\big(\boldsymbol{s} - \eta\mathcal{L}(\boldsymbol{\theta}^*, \boldsymbol{m})\nabla_{\boldsymbol{s}}\ln p(\boldsymbol{m}|\boldsymbol{s})\big), \tag{13}$$

where $\mathcal{P}_{\mathcal{C}}$ denotes an efficient projection mapping and implementation details are provided in the Appendix. After sufficient updates, we observe that most elements of $s$ converge to near $0$ or $1$, which is benefited from the sparsity-inducing feasible region $\mathcal{C}$. Therefore, the randomness of $m$ vanishes and we can derive almost deterministic optimized time steps for sampling with model $\theta^*$.

*Remark* 4.1 (Backpropagation Free.). It is clear that the gradient $\mathcal{L}_{eval}(\theta^*, m)\nabla_s \ln p(m|s)$ requires only forward calculation of $\mathcal{L}_{eval}(\theta^*, m)$ but no backward propagation through the model $\theta^*$. This makes the update of $s$ efficient.

*Remark* 4.2 (Understanding from numerical ODE prespective.). Fundamentally, since our sampling follows the probability flow ODE, optimizing the sampling time steps is equivalent to discretizing the time space to minimize the accumulated error when solved with a specific ODE solver, which is always overlooked by existing efficient samplers.

## 4.2. Inconsistency between Training and Sampling

We analyze the inconsistency between training and sampling via discussing the following two facts:

- The forward process of diffusion models is typically trained across thousands of time steps(e.g., DDPM (Ho et al., 2020) discretizes the time space into 1000 timesteps with uniform training during the training phase) or even continuous time distributions, this process enforces the model to fit the score function at all these time steps. Furthermore, as evidenced by the training objective in Eq. (2), each time step $t$ is trained almost independently, where $x_t = \alpha_t x_0 + \sigma_t \epsilon$ is determined by only the noise level $\alpha_t$ of the current time step and i.i.d random noise $\epsilon$, that is the score function is trained to fit these data sampled independently.

- However, existing sampling acceleration methods based on numerical solvers for probability flow ODE (Song et al., 2021a; Lu et al., 2022a) usually perform inference via calculating the integral curve of noise/score function with merely dozens of selected steps instead of all discretized time steps. This sparse temporal sampling creates a significant gap: the model must now provide outputs at time steps that form a much coarser sequence than during training. Moreover, the numerical integration process implicitly requires these outputs to be consistent and smoothly connected along the trajectory, a requirement not explicitly enforced during the point-wise training phase. As a result, the model may struggle to maintain accuracy and coherence when evaluated under these altered conditions.

The above discrepancy implies that the effort invested in fitting the noise at numerous time steps unused in inference together with the corresponding capacity of the neural network is largely wasted and these efforts should have been used on the selected time steps to improve the performance. Specifically, our experimental results in Table1 substantiate this analysis, where models trained solely on selected time steps outperform that trained on all time steps when conducting skipped sampling. This implies that the model trained on all time steps attends to fit inputs of all noise levels, which is unnecessary and limits its performance at chosen noise levels. The result also justifies the independency of different time steps during training, as canceling steps not used in sampling does not degrade the results, but enhances them instead.

Therefore, we conclude that once upon skipped sampling is adopted, the model should be specially tuned on the selected time steps to ensure the best performance. We thereby put forward the joint training scheme in the next section.

## 4.3. Joint Optimization of Diffusion Model and Timestep Selection

As explained in Sec. 4.2, model performance could be enhanced by specially training or tuning on selected time steps. We also contend that the process of time step selection and model tuning should be performed spontaneously. That is because optimal time steps might differ for distinct models. Therefore, if we select time steps on model $\theta$ and train on selected time steps and derive model $\theta^*$, then optimal time steps for $\theta^*$ might have changed. This naturally leads to a bilevel optimization formulation (Liao et al., 2026). We thus propose the following algorithm:

$$\min_{s} \Phi(s) = \mathbb{E}_{m \sim p(m|s)} \mathcal{L}_{eval}(\theta^*(m), m), \tag{14}$$

$$\theta^*(m) = \arg\min_{\theta} \mathbb{E}_{x_0, \varepsilon, t \sim m} \left[ \omega(t) \|\varepsilon_\theta(x_t, t) - \varepsilon\|_2^2 \right]. \tag{15}$$

The outer loop can be regarded as performing time step selection, while the inner loop is tuning the model based on selected time steps of the current iteration. Thus $s$ should converge to a point where both $s$ and model $\theta$ achieve optimal. The complete algorithm is shown in Alg. 1. It is important to note that our bilevel optimization problem can be solved efficiently. The reasons are as follows. First, in each inner loop training iteration, the model parameters are initialized from the final state of the previous iteration's parameters, which substantially reduces the required training budget for achieving inner loop convergence. Second, our theoretical analysis shows that our algorithm can converge even without fully convergence in the inner loop iterations. The formal justification for this behavior is provided in Theorem 4.3 and Theorem 4.4. (Complete statements and proofs are provided in Appendix B). Hence, in practice, we perform only a fixed small number of inner loop updates each time instead of train the inner loop to convergence.

*Table 1.* FID comparison between full-step training and selected-step retraining.

| Dataset | Config | NFE | | |
|---|---|---|---|---|
| | | 18 | 35 | 256 |
| AFHQV2 | full steps | 8.18 | 3.75 | 2.45 |
| | selected | 6.99 | 3.52 | 2.11 |
| CIFAR10 | full steps | 7.41 | 3.43 | 2.03 |
| | selected | 6.15 | 3.09 | 1.84 |

*Table 2.* FID comparison under different timestep schedules.

| Solver | Time Sche. | NFE | | |
|---|---|---|---|---|
| | | 8 | 9 | 10 |
| iPNDM | EDM | 5.55 | 5.00 | 3.89 |
| | LogSNR | 5.73 | 5.21 | 4.39 |
| Uni-PC | EDM | 9.65 | 7.83 | 6.12 |
| | LogSNR | 4.41 | 3.55 | 3.16 |

---

**Algorithm 1** PCS with Fine-Tuning

---

1: **Input:** Initial parameters $s_0, \theta_0$, inner step size $\alpha$, outer step sizes $\{\eta_t\}$, number of inner steps $k$
2: **Output:** Optimized $s$
3: **for** $t = 0, 1, 2, \ldots$ **do**
4: $\quad$ Sample $m_t \sim p(m|s_t)$
5: $\quad$ Set $\theta_{t,0} = \theta_t$
6: $\quad$ **for** $i = 1, 2, \ldots, k$ **do**
7: $\quad\quad$ Sample minibatch $x_i \sim \mathcal{D}$
8: $\quad\quad$ $\theta_{t,i} = \theta_{t,i-1} - \alpha \nabla_\theta g(\theta_{t,i-1}(x_i), m_t)$
9: $\quad$ **end for**
10: $\quad$ Set $\theta_{t+1} = \theta_{t,k}$
11: $\quad$ Compute approximate gradient: $\nabla_s \Phi_{\text{approx}}(s_t) = f(\theta_{t,k}, m_t)\nabla_s \ln p(m_t|s_t)$
12: $\quad$ Update $s$: $s_{t+1} = s_t - \eta_t \nabla_s \Phi_{\text{approx}}(s_t)$
13: **end for**

---

**Theorem 4.3.** (Convergence of $\theta$) [Informal] *Under proper assumptions on inner loop loss function $\mathcal{L}$, distance of neighboring masks $m_t$ and $m_{t+1}$, outer and inner learning rate $\eta$ and $\alpha$, we have:* $\mathbb{E}\big[\|\theta_{t,k} - \theta^*(m_t))\|\big] \leq \sqrt{C_1^{k(t-t_0)}} \, \mathbb{E}\big[\|\theta_0 - \theta^*(m_0))\|\big] + C_2$, *where $\theta_{t,k}$ represents $\theta$ at $k$-th inner iteration of $t$-th outer iteration. $C_1 < 1$ and $C_2$ are small constants related to the assumptions.*

**Theorem 4.4.** (Convergence of $\Phi(s)$) [Informal] *Under assumptions in* Theorem *4.3 and proper assumptions on $\Phi(s)$ and $p(m|s)$, we have:*

$$\mathbb{E}[\Phi(s_{t+1})] \leq \mathbb{E}[\Phi(s_t)].$$

## 5. Experiments

In this section, we present our experimental framework and results. Section 5.1 first introduces the experimental setup, with additional details available in the supplementary materials. Subsequently, Section 5.2 provides comprehensive comparisons between our method and both manually-designed approaches and various optimization-based timestep selection methods, accompanied by visualizations of the results. Finally, Section 5.3 presents a sensitivity analysis of the key hyperparameters in our algorithm.

### 5.1. Experiments Setup

**Datasets and Evaluation Metric.** We conduct extensive experiments on pixel-space diffusion models to evaluate our method. Following standard benchmarks in the field, we employ four datasets: CIFAR-10 (Krizhevsky et al., 2009) (32×32), FFHQ-64×64 (Karras et al., 2019), AFHQv2-64×64 (Choi et al., 2020), and ImageNet-64×64 (Russakovsky et al., 2015), covering diverse image domains from natural scenes to facial and animal images. The Fréchet Inception Distance (FID) (Heusel et al., 2017) is adopted as the primary metric for quantitative comparison. To provide deeper insights, we include comprehensive visualizations of generated samples under different sampling steps.

**Baselines and Sampling Setup.** We extensively compare our method with a wide range of baseline approaches, including both handcrafted and optimization-based discrete timestep schedules. The handcrafted baselines consist of EDM (Karras et al., 2022), uniform (Ho et al., 2020), and logSNR (Zhang & Chen, 2022) schedules(more details refer to Appendix A), while the optimization-based counterparts include DMN (Xue et al., 2024), GITs (Chen et al., 2024), and LD3 (Tong et al., 2024). For sampling, we primarily employ two classical samplers: DDIM (Song et al., 2021a), iPNDM (Liu et al., 2022), under the constraint of 4–10 NFEs to ensure fair and efficient comparison.

**Training Details.** For the main experiments in Table 3, we employ the NCSN++ (Song et al., 2021b) model architecture within the EDM (Karras et al., 2022) framework. The candidate set of timesteps for the outer loop is defaulted to 128, with initial timestep selection following the hand-designed rules of EDM (see Appendix A for details). The outer loop learning rate is set to 1e-2, while the inner loop uses a learning rate of 1e-5.The candidate set of timesteps for the outer loop defaults to 128, with these timesteps selected according to the hand-designed rules of EDM. Building upon this, we initialize each timestep with uniform spacing, specifically setting $s_i = (K - 2)/128$(we found this initialization already achieves sufficiently good performance). The outer loop utilizes 50K images for forward computation

*Table 3.* FID comparison on CIFAR10, AFHQv2, Imagenet and FFHQ and the two solvers DDIM and iPNDM. We compare PCS with three different time discretization optimization methods, DMN,GITS and LD3. FID scores are computed with 50k samples using the reference data set. **Bold** indicates the best result, and underline indicates the second-best.

| Method | NFE | | | | Method | NFE | | | |
|---|---|---|---|---|---|---|---|---|---|
| | 4 | 6 | 8 | 10 | | 4 | 6 | 8 | 10 |
| **CIFAR10 32x32(Krizhevsky et al., 2009)** | | | | | **FFHQ 64x64 (Karras et al., 2019)** | | | | |
| DDIM(GITS) | 42.86 | 21.04 | 13.30 | 10.37 | DDIM(GITS) | 40.80 | 23.67 | 16.78 | 13.06 |
| DDIM(DMN) | 37.64 | 23.03 | 14.96 | 12.17 | DDIM(DMN) | 35.69 | 22.15 | 15.96 | 14.45 |
| DDIM(LD3) | 26.04 | 18.98 | 12.96 | 9.61 | DDIM(LD3) | 29.07 | 22.01 | 13.07 | 11.23 |
| DDIM(PCS)(**ours**) | 25.22 | 19.08 | 11.79 | 9.91 | DDIM(PCS)(**ours**) | 26.19 | 20.66 | 12.54 | 10.88 |
| DDIM(PCS+ft)(**ours**) | **21.09** | **17.77** | **9.54** | **9.25** | DDIM(PCS+ft)(**ours**) | **23.95** | **17.25** | **11.02** | **10.45** |
| iPNDM(GITS) | 15.63 | 6.82 | 4.29 | 2.78 | iPNDM(GITS) | 17.03 | 7.00 | 4.53 | 3.62 |
| iPNDM(DMN) | 28.09 | 9.24 | 7.68 | 3.31 | iPNDM(DMN) | 19.69 | 6.88 | 5.04 | 4.29 |
| iPNDM(LD3) | 9.31 | 3.35 | 2.81 | 2.38 | iPNDM(LD3) | 17.96 | 6.47 | 3.97 | 3.25 |
| iPNDM(PCS)(**ours**) | 8.96 | 3.12 | 2.73 | 2.29 | iPNDM(PCS)(**ours**) | 15.28 | 6.33 | 4.06 | 3.20 |
| iPNDM(PCS+ft)(**ours**) | **8.75** | **3.01** | **2.55** | **2.14** | iPNDM(PCS+ft)(**ours**) | **14.97** | **5.87** | **3.72** | **3.03** |
| **Imagenet 64x64 (Russakovsky et al., 2015)** | | | | | **AFHQv2 64x64 (Choi et al., 2020)** | | | | |
| DDIM(GITS) | 33.42 | 18.51 | 12.79 | 10.83 | DDIM(GITS) | 23.50 | 14.38 | 10.38 | 7.61 |
| DDIM(DMN) | 30.71 | 25.93 | 15.26 | 12.97 | DDIM(DMN) | 26.36 | 15.42 | 11.81 | 7.73 |
| DDIM(LD3) | 22.18 | 16.45 | 12.07 | 10.03 | DDIM(LD3) | 20.54 | 12.67 | 10.02 | 7.46 |
| DDIM(PCS)(**ours**) | 20.98 | 14.18 | 11.16 | 9.69 | DDIM(PCS)(**ours**) | 21.32 | 11.43 | 9.40 | 7.67 |
| DDIM(PCS+ft)(**ours**) | **18.27** | **13.09** | **10.54** | **9.25** | DDIM(PCS+ft)(**ours**) | **18.66** | **10.73** | **9.05** | **6.45** |
| iPNDM(GITS) | 17.56 | 8.43 | 5.82 | 4.48 | iPNDM(GITS) | 13.25 | 5.48 | 3.42 | 2.97 |
| iPNDM(DMN) | 20.14 | 11.36 | 6.36 | 4.96 | iPNDM(DMN) | 17.91 | 7.48 | 3.87 | 3.12 |
| iPNDM(LD3) | 15.69 | 8.01 | 5.26 | 4.10 | iPNDM(LD3) | 9.96 | 3.63 | 2.63 | 2.27 |
| iPNDM(PCS)(**ours**) | 14.97 | 7.54 | 5.11 | 3.93 | iPNDM(PCS)(**ours**) | 10.15 | 3.77 | 2.45 | 2.19 |
| iPNDM(PCS)(**ours**) | **11.75** | **7.02** | **4.88** | **3.70** | iPNDM(PCS+ft)(**ours**) | **8.98** | **3.40** | **2.23** | **2.01** |

of FID scores to compute the loss function for model performance evaluation, while the inner loop performs 1K steps of fine-tuning on the complete dataset. Both loops employ a batch size of 128. All experiments were conducted using 4 NVIDIA A100 GPUs with 80GB memory each.

## 5.2. Main Results

**Performance Comparison with Optimization-Based Methods.** Table 3 presents a comparison between our proposed method and several optimization-based approaches on CIFAR10, ImageNet, FFHQ, and AFHQv2. The results demonstrate that our method exhibits strong competitiveness against all baselines in the NFE range of 4–10. Furthermore, our bilevel optimization framework(i.e. PCS with fine-tuning) consistently delivers additional performance improvements. The performance unlocked by fine-tuning validates our analysis about inconsistency between training and sampling. It is also noteworthy that fine-tuning leads to more significant performance improvements when the NFE is small. This may be attributed to the fact that the training–sampling inconsistency becomes more severe with fewer sampling steps.

**Comparison with Handcrafted Timestep Rules.** As shown in Table 4, we compare our method with several manually designed timestep scheduling rules under DDIM and iPNDM solvers. The results clearly indicate that handcrafted

rules are generally suboptimal, whereas our optimization algorithm significantly improves generation quality under the same NFE budget. Additional visual comparisons of generated samples are provided in Figure 1a and Figure 1b.

*Table 4.* FID on CIFAR10 with heuristic schedules (DDIM/iPNDM).

| Time Sch. | NFE | | | |
|---|---|---|---|---|
| | 4 | 6 | 8 | 10 |
| DDIM(uniform) | 83.10 | 44.70 | 26.13 | 17.62 |
| DDIM(logsnr) | 73.19 | 38.19 | 24.05 | 16.43 |
| DDIM(EDM) | 66.76 | 35.61 | 22.31 | 15.68 |
| **DDIM(PCS)** | **25.22** | **19.08** | **11.79** | **9.91** |
| iPNDM(uniform) | 58.16 | 29.18 | 12.85 | 6.58 |
| iPNDM(logsnr) | 35.58 | 11.92 | 5.73 | 4.39 |
| iPNDM(EDM) | 29.98 | 10.02 | 5.55 | 3.89 |
| **iPNDM(PCS)** | **8.96** | **3.12** | **2.73** | **2.29** |

## 5.3. Ablation Study

In this section, we present a series of experiments designed to evaluate the influence of various hyperparameter settings on our method, e.g., candidate set size $N$ and the initialization strategy for candidate set.

| Uniform | LogSNR | EDM | PCS |
|---|---|---|---|

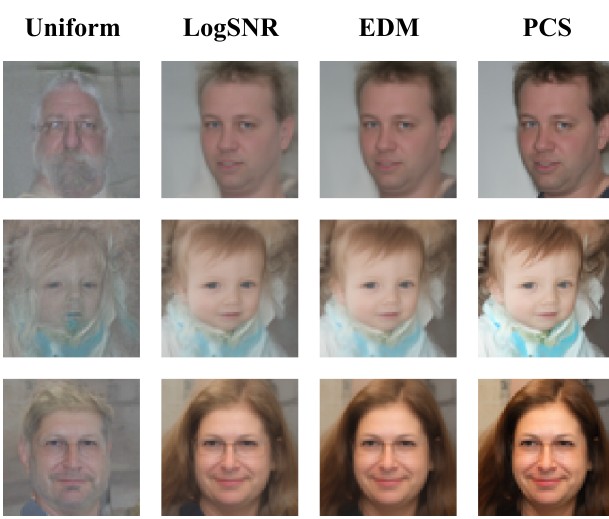

*(a)* FFHQ–DDIM sampler with NFE=7.

| Uniform | LogSNR | EDM | PCS |
|---|---|---|---|

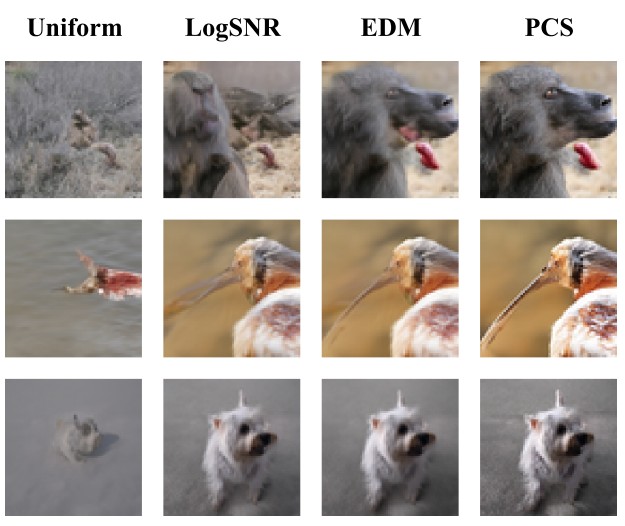

*(b)* ImageNet–iPNDM sampler with NFE=7.

*(c)* Sample visualizations of PCS under DDIM (top) and iPNDM (bottom).

**Candidate Set Size** $N$**.** We evaluate different candidate set sizes $N$ selected based on the hand-crafted rules from EDM (Karras et al., 2022) (see Appendix A for details). Experimental results in Tab5 demonstrate that increasing the candidate set size enables our algorithm to discover superior sampling schedules. DDIM(128) achieves the best FID across all NFEs, while reducing the candidate size leads to clear degradation, especially at low NFEs (e.g., 25.22 vs. 29.76 at NFE=4). This trend indicates that larger candidate pools provide more informative samples for PCS, resulting in better discretization and higher-quality generation.

**Initialization Strategy.** The initialization strategy proves equally crucial. In Table6, sophisticated hand-designed rules like EDM (Karras et al., 2022) initialization for con-

*Table 5.* PCS with DDIM on CIFAR10: candidate set size.

| Cand Size. | NFE | | | |
|---|---|---|---|---|
| | 4 | 6 | 8 | 10 |
| DDIM(128) | **25.22** | **19.08** | **11.79** | **9.91** |
| DDIM(64) | 29.33 | 21.17 | 14.01 | 10.68 |
| DDIM(32) | 29.76 | 23.24 | 14.98 | 12.35 |

structing the timestep candidate set consistently outperform naive uniform sampling, achieving better final performance metrics.

*Table 6.* PCS with DDIM on CIFAR10: initialization strategy.

| Init Strategy. | NFE | | | |
|---|---|---|---|---|
| | 4 | 6 | 8 | 10 |
| Uniform | 35.24 | 20.66 | 15.98 | 12.62 |
| LogSNR | 29.87 | **17.65** | 13.09 | 10.87 |
| EDM | **25.22** | 19.08 | **11.79** | **9.91** |

As shown in Table 6, the choice of initialization strategy has a significant impact on PCS performance. Uniform sampling yields the weakest results across all NFEs, while both LogSNR and EDM provide substantial improvements. LogSNR shows strong performance at mid-range NFEs (e.g., **17.65** at NFE=6), but EDM achieves the best or near-best FID in most settings, particularly at low and high NFEs (e.g., **25.22** at NFE=4 and **9.91** at NFE=10). These results suggest that better-aligned timestep priors offer more effective candidate sets for PCS optimization, leading to consistently higher sampling quality.

## 6. Conclusion

In this work, we propose a policy gradient-based optimization method for timestep selection in diffusion model sampling. By introducing a probabilistic mask reparameterization technique, we transform the originally discrete and non-differentiable problem into a continuous and differentiable optimization framework. Furthermore, we identify and address the training-inference discrepancy inherent in diffusion models by extending our approach to a bilevel optimization formulation, which effectively mitigates this inconsistency. We provide theoretical convergence guarantees for the proposed algorithm and demonstrate its effectiveness through extensive experimental validation. Our results show consistent improvements over conventional sampling approaches across multiple benchmarks.

## Impact Statement

This paper presents work whose goal is to improve the efficiency of diffusion model sampling. Potential societal impacts are primarily those associated with generative modeling systems; we do not identify additional impacts that require special discussion beyond the established considerations for this area.

## Acknowledgements

This work was supported by the National Nature Science Foundation of China (62472097), Shanghai Municipal Science and Technology Commission (Grant No.24511106102), AI for Science Foundation of Fudan University (FudanX24AI028) and Fudan Kunpeng&Ascend Center of Cultivation. The computations in this research were performed on the CFFF platform of Fudan University. We also thank Yuquan Zhou for helpful assistance during the rebuttal period.

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

# Supplemental Material: Efficient Diffusion Models via Time Step Optimization with Consistent Training and Inference Constraints

This appendix can be divided into 4 parts. To be precise,

1. Section A supplements additional experimental details.

2. Section B provides the detailed proof of Theorem 4.3, and Theorem 4.4.

3. Section C discusses the limitation of this paper.

4. Section D discusses the boader impacts of this paper.

## A. Additional Experimental setup

We compare our learned time discretization with the following discretizations heuristics:

**Polynomial discretization (Time uniform):** This discretization is a polynomial function. Specifically:

$$t_i = \left(\frac{i}{N}\right)^\rho (t_{\max} - t_{\min}) + t_{\min}, \quad t_{\max} = T, \ i = 0, 1, \ldots, N. \tag{16}$$

**Time EDM discretization:** Time EDM discretization (Karras et al., 2022) has been shown to be effective with Heun's solver on EDM pre-trained model:

$$\sigma(t_i) = \left(\sigma_{max}^{-\rho} + \frac{i}{N-1}\left(\sigma_{min}^{-\rho} - \sigma_{max}^{-\rho}\right)\right)^\rho. \tag{17}$$

**Time LogSNR:** This schedule uniformly separate the logSNR from (Lu et al., 2022a). Specifically:

$$\lambda(t_i) = \frac{N-i}{N}(\lambda_{\max} - \lambda_{\min}) + \lambda_{\min}, \quad \text{where } \lambda(t_i) = \frac{\alpha(t_i)}{\sigma(t_i)}. \tag{18}$$

**Initialization of candidate set.** Specifically, we initialize the candidate set using handcrafted rules from EDM. For instance, when the candidate set size is 128, we set $N = 128$ in Equation 17, which yields 128 initial points. If our target NFE (Number of Function Evaluations) is 7, we assign each of these 128 points an initial $s$ value of $7/128$.

## B. Proofs

In the context of deep learning where complete theoretical characterization remains challenging, the subsequent analysis establishes convergence properties under idealized conditions. These results should be interpreted as supporting evidence rather than exhaustive theoretical justification, as the core contribution of our work is principally algorithmic.

**Lemma B.1.** *Let $h(\boldsymbol{\theta}) = \mathbb{E}_x\left[g(\boldsymbol{\theta}; \boldsymbol{x})\right]$ be $\mu$-**strongly convex** and **$L$-smooth**. Assume the variance of the stochastic gradient $\nabla_\theta g(\boldsymbol{\theta}; \boldsymbol{x})$ is upper bounded by $\sigma^2$. Then, the convergence of $\boldsymbol{\theta}$ under SGD update $\boldsymbol{\theta_k} \leftarrow \boldsymbol{\theta_{k-1}} - \alpha\nabla_\theta g(\boldsymbol{\theta_{k-1}}; \boldsymbol{x})$ is:*

$$\mathbb{E}\left[\|\boldsymbol{\theta_k} - \boldsymbol{\theta^*}\|^2\right] \leq (1 - \mu\alpha)^k \|\boldsymbol{\theta_0} - \boldsymbol{\theta^*}\|^2 + \frac{\alpha\sigma^2}{\mu}, \tag{19}$$

*where $\boldsymbol{\theta_k}$ is the value of $\boldsymbol{\theta}$ after $k$ iterations, $\alpha \leq \frac{\mu}{L^2}$ is the learning rate, $\boldsymbol{\theta^*}$ represents the model parameters converged on the given training dataset. This is in fact a classical result in stochastic gradient descent (SGD) theory, which we include here for the sake of self-containedness in our presentation.*

*Proof.* Denote the stochastic gradient at step $i$ as :

$$\tilde{\nabla}h(\boldsymbol{\theta_{i-1}}) = \nabla_\theta g(\boldsymbol{\theta_{i-1}}; \boldsymbol{x_i}).$$

This is an unbiased estimator, i.e.,

$$\mathbb{E}_{x_i}[\tilde{\nabla}h(\boldsymbol{\theta_{i-1}})] = \nabla h(\boldsymbol{\theta_{i-1}}).$$

Let:

$$\tilde{\nabla}h(\boldsymbol{\theta_{i-1}}) = \nabla h(\boldsymbol{\theta_{i-1}}) + \xi_i,$$

where $\xi_i = \tilde{\nabla}h(\boldsymbol{\theta_{i-1}}) - \nabla h(\boldsymbol{\theta_{i-1}})$ is the noise term, with $\mathbb{E}_{x_i}[\xi_i] = 0$ and $\mathbb{E}_{x_i}[\|\xi_i^2\|] \leq \sigma^2$. Then the SGD update becomes:

$$\boldsymbol{\theta_i} = \boldsymbol{\theta_{i-1}} - \alpha\tilde{\nabla}h(\boldsymbol{\theta_{i-1}}) = \boldsymbol{\theta_{i-1}} - \alpha\nabla h(\boldsymbol{\theta_{i-1}}) - \alpha\xi_i.$$

Therefore, the squared distance to $\boldsymbol{\theta^*}$ is:

$$\|\boldsymbol{\theta_i} - \boldsymbol{\theta^*}\|^2 = \|(\boldsymbol{\theta_{i-1}} - \boldsymbol{\theta^*}) - \alpha\nabla h(\boldsymbol{\theta_{i-1}}) - \alpha\xi_i\|^2$$
$$= \|(\boldsymbol{\theta_{i-1}} - \boldsymbol{\theta^*}) - \alpha\nabla h(\boldsymbol{\theta_{i-1}})\|^2 - 2\alpha\langle(\boldsymbol{\theta_{i-1}} - \boldsymbol{\theta^*}) - \alpha\nabla h(\boldsymbol{\theta_{i-1}}), \xi_i\rangle + \alpha^2\|\xi_i\|^2. \quad (20)$$

Taking expectation on both sides with respect to $x_i$ conditioning on $\theta_{i-1}$:

$$\mathbb{E}_{x_i}[\|\boldsymbol{\theta_i} - \boldsymbol{\theta^*}\|^2|\boldsymbol{\theta_{i-1}}] = \|(\boldsymbol{\theta_{i-1}} - \boldsymbol{\theta^*}) - \alpha\nabla h(\boldsymbol{\theta_{i-1}})\|^2 - 2\alpha\mathbb{E}_{x_i}[\langle(\boldsymbol{\theta_{i-1}} - \boldsymbol{\theta^*}) - \alpha\nabla h(\boldsymbol{\theta_{i-1}}), \xi_i\rangle]$$
$$+ \alpha^2\mathbb{E}_{x_i}[\|\xi_i\|^2]$$
$$= \|(\boldsymbol{\theta_{i-1}} - \boldsymbol{\theta^*}) - \alpha\nabla h(\boldsymbol{\theta_{i-1}})\|^2 + \alpha^2\mathbb{E}_{x_i}[\|\xi_i\|^2]$$
$$\leq \|(\boldsymbol{\theta_{i-1}} - \boldsymbol{\theta^*}) - \alpha\nabla h(\boldsymbol{\theta_{i-1}})\|^2 + \alpha^2\sigma^2. \quad (21)$$

The second equation is because $\mathbb{E}_{x_I}[\|\xi_i\|] = 0$.

Next, bound the term $\|(\boldsymbol{\theta_{i-1}} - \boldsymbol{\theta^*}) - \alpha\nabla h(\boldsymbol{\theta_{i-1}})\|^2$. By strong convexity of $h(\boldsymbol{\theta})$ we have:

$$h(\boldsymbol{\theta^*}) \geq h(\boldsymbol{\theta}) + \langle\nabla h(\boldsymbol{\theta}), \boldsymbol{\theta^*} - \boldsymbol{\theta}\rangle + \frac{\mu}{2}\|\boldsymbol{\theta} - \boldsymbol{\theta^*}\|^2, \quad (22)$$

$$h(\boldsymbol{\theta}) \geq h(\boldsymbol{\theta^*}) + \langle\nabla h(\boldsymbol{\theta^*}), \boldsymbol{\theta} - \boldsymbol{\theta^*}\rangle + \frac{\mu}{2}\|\boldsymbol{\theta} - \boldsymbol{\theta^*}\|^2 = h(\boldsymbol{\theta^*}) + \frac{\mu}{2}\|\boldsymbol{\theta} - \boldsymbol{\theta^*}\|^2. \quad (23)$$

Adding Eq. (22) and 23 together we have:

$$\langle\nabla h(\boldsymbol{\theta_{i-1}}), \boldsymbol{\theta_{i-1}} - \boldsymbol{\theta^*}\rangle \geq \mu\|\boldsymbol{\theta_{i-1}} - \boldsymbol{\theta^*}\|^2.$$

Since $h$ is L-smooth, the gradient satisfies:

$$\|\nabla h(\boldsymbol{\theta_{i-1}}) - \nabla h(\boldsymbol{\theta^*})\|^2 \leq L^2\|\boldsymbol{\theta_{i-1}} - \boldsymbol{\theta^*}\|^2.$$

Combine these together, we have:

$$\|(\boldsymbol{\theta_{i-1}} - \boldsymbol{\theta^*}) - \alpha\nabla h(\boldsymbol{\theta_{i-1}})\|^2 \quad (24)$$
$$\leq \|\boldsymbol{\theta_{i-1}} - \boldsymbol{\theta^*}\|^2 - 2\alpha\mu\|\boldsymbol{\theta_{i-1}} - \boldsymbol{\theta^*}\|^2 + \alpha^2 L^2\|\boldsymbol{\theta_{i-1}} - \boldsymbol{\theta^*}\|^2$$
$$= (1 - 2\alpha\mu + \alpha^2 L^2)\|\boldsymbol{\theta_{i-1}} - \boldsymbol{\theta^*}\|^2$$
$$= [1 - \alpha\mu + \alpha L^2(\alpha - \frac{\mu}{L^2})]\|\boldsymbol{\theta_{i-1}} - \boldsymbol{\theta^*}\|^2$$
$$\leq (1 - \alpha\mu)\|\boldsymbol{\theta_{i-1}} - \boldsymbol{\theta^*}\|^2 \quad (25)$$

Substitute back to Eq. (21), we have:

$$\mathbb{E}_{x_i}[\|\boldsymbol{\theta_i} - \boldsymbol{\theta^*}\|^2|\boldsymbol{\theta_{i-1}}] \leq (1 - \mu\alpha)\|\boldsymbol{\theta_{i-1}} - \boldsymbol{\theta^*}\|^2 + \alpha^2\sigma^2.$$

Taking expectation on $\boldsymbol{\theta_{i-1}}$:

$$\mathbb{E}[\|\boldsymbol{\theta_i} - \boldsymbol{\theta^*}\|^2] = \mathbb{E}[\mathbb{E}_{x_i}[\|\boldsymbol{\theta_i} - \boldsymbol{\theta^*}\|^2|\boldsymbol{\theta_{i-1}}]] \leq (1 - \mu\alpha)\mathbb{E}[\|\boldsymbol{\theta_{i-1}} - \boldsymbol{\theta^*}\|^2] + \alpha^2\sigma^2$$

Therefore, at $k$-th iteration, we have:

$$\mathbb{E}[\|\boldsymbol{\theta_k} - \boldsymbol{\theta^*}\|^2] \leq (1 - \mu\alpha)^k\mathbb{E}[\|\boldsymbol{\theta_0} - \boldsymbol{\theta^*}\|^2] + \frac{\alpha\sigma^2}{\mu}(1 - (1 - \mu\alpha)^k)$$

$$\leq (1 - \mu\alpha)^k\|\boldsymbol{\theta_0} - \boldsymbol{\theta^*}\|^2 + \frac{\alpha\sigma^2}{\mu} \quad (26)$$

$$\square$$

**Theorem B.2** (Convergence of $\boldsymbol{\theta}$). *For the following bi-level optimization problem:*

$$\min_{\boldsymbol{s}} \Phi(\boldsymbol{s}) = \mathbb{E}_{\boldsymbol{m}\sim p(\boldsymbol{m}|\boldsymbol{s})}\mathcal{L}_{outer}(\boldsymbol{\theta}^*(\boldsymbol{m}), \boldsymbol{m}) \tag{27}$$

$$s.t.\ \boldsymbol{\theta}^*(\boldsymbol{m}) \in \arg\min_{\boldsymbol{\theta}} \mathbb{E}_x \mathcal{L}_{inner}(\boldsymbol{\theta}, \boldsymbol{x}, \boldsymbol{m}).$$

*Let $h(\boldsymbol{\theta}) = \mathbb{E}_x \mathcal{L}_{inner}(\boldsymbol{\theta}, \boldsymbol{x}, \boldsymbol{m})$. Assume $h(\boldsymbol{\theta})$ is L-smooth in $\theta$ (i.e. $\nabla_\theta h(\boldsymbol{\theta})$ is L-Lipschitz) and $\mu$-strongly convex in $\theta$. Assume further that the variance of the norm of the inner stochastic gradient $\|\nabla_{\boldsymbol{\theta}}\mathcal{L}_{inner}(\boldsymbol{\theta}, \boldsymbol{x}, \boldsymbol{m})\|$ is upper bounded by $\sigma^2$. Let $D = \max_{t=1,2,...,T} \mathbb{E}\big[\|\boldsymbol{\theta}^*(\boldsymbol{m}_t) - \boldsymbol{\theta}^*(\boldsymbol{m}_{t+1})\|\big]$). Then, $\mathbb{E}\big[\|\boldsymbol{\theta}_{t,k} - \boldsymbol{\theta}^*(\boldsymbol{m}_t))\|\big] \leq \sqrt{(1-\mu\alpha)^{k(t-t_0)}}\, \mathbb{E}\big[\|\boldsymbol{\theta}_0 - \boldsymbol{\theta}^*(\boldsymbol{m}_0))\|\big] + \big(\sqrt{\frac{\alpha\sigma^2}{\mu}} + D\big)$, i.e., $\boldsymbol{\theta}$ converges toward inner loop optimal.*

*Proof.* By Lemma B.1 we have:

$$\mathbb{E}\left[\|\theta_{t,k} - \boldsymbol{\theta}^*(\boldsymbol{m}_t))\|^2\right] \leq (1-\mu\alpha)^k \|\boldsymbol{\theta}_t - \boldsymbol{\theta}^*(\boldsymbol{m}_t)\|^2 + \frac{\alpha\sigma^2}{\mu}, \tag{28}$$

where $\boldsymbol{m}_t$ is the masked generated from $p(\boldsymbol{m}|\boldsymbol{s})$, $\boldsymbol{\theta}_t$ is the model weights at $t$-th outer iteration, and $\boldsymbol{\theta}_{t_k}$ is that at $k$-th inner iteration of $t$-th outer iteration.

Let $t = t - 1$ and by Jensen's inequality:

$$\mathbb{E}\left[\|\boldsymbol{\theta}_{t-1,k} - \boldsymbol{\theta}^*(\boldsymbol{m}_{t-1}))\|\big|\boldsymbol{\theta}_{t-1}\right] \leq \sqrt{\mathbb{E}\left[\|\boldsymbol{\theta}_{t-1,k} - \boldsymbol{\theta}^*(\boldsymbol{m}_{t-1}))\|^2\big|\boldsymbol{\theta}_{t-1}\right]}$$

$$\leq \sqrt{(1-\mu\alpha)^k}\|\boldsymbol{\theta}_{t-1} - \boldsymbol{\theta}^*(\boldsymbol{m}_{t-1}))\| + \sqrt{\frac{\alpha\sigma^2}{\mu}} \tag{29}$$

Also note that:

$$\|\boldsymbol{\theta}_t - \boldsymbol{\theta}^*(\boldsymbol{m}_t)\| = \|\boldsymbol{\theta}_{t-1,k} - \boldsymbol{\theta}^*(\boldsymbol{m}_t)\| \leq \|\boldsymbol{\theta}_{t-1,k} - \boldsymbol{\theta}^*(\boldsymbol{m}_{t-1})\| + \|\boldsymbol{\theta}^*(\boldsymbol{m}_{t-1}) - \boldsymbol{\theta}^*(\boldsymbol{m}_t)\| \tag{30}$$

Taking expectation on both sides conditioning on $\boldsymbol{\theta}_{t-1}$ yields:

$$\mathbb{E}\big[\|\boldsymbol{\theta}_t - \boldsymbol{\theta}^*(\boldsymbol{m}_t)\|\big|\boldsymbol{\theta}_{t-1}\big] \leq \mathbb{E}\big[\|\boldsymbol{\theta}_{t-1,k} - \boldsymbol{\theta}^*(\boldsymbol{m}_{t-1})\|\big|\boldsymbol{\theta}_{t-1}\big] + \|\boldsymbol{\theta}^*(\boldsymbol{m}_{t-1}) - \boldsymbol{\theta}^*(\boldsymbol{m}_t)\| \tag{31}$$

Plug Eq. (29) into Eq. (31) we have:

$$\mathbb{E}\big[\|\boldsymbol{\theta}_{t,k} - \boldsymbol{\theta}^*(\boldsymbol{m}_t))\|\big|\boldsymbol{\theta}_t\big] \leq \sqrt{(1-\mu\alpha)^k}\|\boldsymbol{\theta}_{t-1} - \boldsymbol{\theta}^*(\boldsymbol{m}_{t-1}))\| + \sqrt{\frac{\alpha\sigma^2}{\mu}} + \|\boldsymbol{\theta}^*(\boldsymbol{m}_{t-1}) - \boldsymbol{\theta}^*(\boldsymbol{m}_t)\| \tag{32}$$

Taking expectation on both sides with respect to $\boldsymbol{\theta}_{t-1}$ yields:

$$\mathbb{E}\big[\|\boldsymbol{\theta}_{t,k}-\boldsymbol{\theta}^*(\boldsymbol{m}_t))\|\big] \leq \sqrt{(1-\mu\alpha)^k}\, \mathbb{E}\big[\|\boldsymbol{\theta}_{t-1} - \boldsymbol{\theta}^*(\boldsymbol{m}_{t-1}))\|\big] + \sqrt{\frac{\alpha\sigma^2}{\mu}} + \mathbb{E}\big[\|\boldsymbol{\theta}^*(\boldsymbol{m}_{t-1}) - \boldsymbol{\theta}^*(\boldsymbol{m}_t)\|\big]$$

$$\leq \sqrt{(1-\mu\alpha)^k}\, \mathbb{E}\big[\|\boldsymbol{\theta}_{t-1} - \boldsymbol{\theta}^*(\boldsymbol{m}_{t-1}))\|\big] + \sqrt{\frac{\alpha\sigma^2}{\mu}} + \mathbb{E}\big[\|\boldsymbol{\theta}^*(\boldsymbol{m}_{t-1}) - \boldsymbol{\theta}^*(\boldsymbol{m}_t)\|\big]$$

$$\leq \sqrt{(1-\mu\alpha)^k}\, \mathbb{E}\big[\|\boldsymbol{\theta}_{t-1} - \boldsymbol{\theta}^*(\boldsymbol{m}_{t-1}))\|\big] + \sqrt{\frac{\alpha\sigma^2}{\mu}} + \mathbb{E}\big[\|\boldsymbol{\theta}^*(\boldsymbol{m}_{t-1}) - \boldsymbol{\theta}^*(\boldsymbol{m}_t)\|\big]$$

$$\leq \sqrt{(1-\mu\alpha)^k}\, \mathbb{E}\big[\|\boldsymbol{\theta}_{t-1} - \boldsymbol{\theta}^*(\boldsymbol{m}_{t-1}))\|\big] + \sqrt{\frac{\alpha\sigma^2}{\mu}} + \mathbb{E}\big[\|\boldsymbol{\theta}^*(\boldsymbol{m}_{t-1}) - \boldsymbol{\theta}^*(\boldsymbol{m}_t)\|\big]$$

$$= \sqrt{(1-\mu\alpha)^{k(t-t_0)}}\, \mathbb{E}\big[\|\boldsymbol{\theta}_0 - \boldsymbol{\theta}^*(\boldsymbol{m}_0))\|\big] + (1 - \sqrt{(1-\mu\alpha)^{k(t-t_0)}})\big(\sqrt{\frac{\alpha\sigma^2}{\mu}} +$$

$$\mathbb{E}\big[\|\boldsymbol{\theta}^*(\boldsymbol{m}_{t-1}) - \boldsymbol{\theta}^*(\boldsymbol{m}_t)\|\big]\big)$$

$$\leq \sqrt{(1-\mu\alpha)^{k(t-t_0)}}\, \mathbb{E}\big[\|\boldsymbol{\theta}_0 - \boldsymbol{\theta}^*(\boldsymbol{m}_0))\|\big] + \big(\sqrt{\frac{\alpha\sigma^2}{\mu}} + D\big), \tag{33}$$

Therefore, the model weight $\boldsymbol{\theta}$ converges with a bias related with inner and outer learning rates $\eta$ and $\alpha$, the gradient $\nabla_{\boldsymbol{s}}\Phi(\boldsymbol{s}_t)$, and the upper bound of the distance between neighbor mask $\boldsymbol{m}_{t-1}$ and $\boldsymbol{m}_t$.

$\square$

**Theorem B.3** (Convergence of $\Phi(\boldsymbol{s})$). *Assume all conditions in* Theorem *4.3 hold. Assume further that $f(\boldsymbol{\theta}, \boldsymbol{m})$ is $K$-Lipschitz in $\boldsymbol{\theta}$ for $\forall \boldsymbol{m}$, $\Phi(\boldsymbol{s})$ is $L_s$-smooth in $\boldsymbol{s}$ and $\|\nabla_{\boldsymbol{s}}\Phi(\boldsymbol{s})\|$ is upper bounded, and $\nabla_s \ln p(\boldsymbol{m}|\boldsymbol{s})$ is bounded: $\|\nabla_{\boldsymbol{s}} \ln p(\boldsymbol{m}|\boldsymbol{s})\| \leq B$ for all $\boldsymbol{m}, \boldsymbol{s}$. Then with small outer and inner learning rates $\eta_t$ and $\alpha$, and sufficient inner iteration $k$, we have: $\mathbb{E}[\Phi(\boldsymbol{s}_{t+1})] \leq \mathbb{E}[\Phi(\boldsymbol{s}_t)]$.*

*Proof.* The true gradient is:

$$\nabla_{\boldsymbol{s}}\Phi(\boldsymbol{s}_t) = \mathbb{E}_{\boldsymbol{m}\sim p(\boldsymbol{m}|\boldsymbol{s}_t)}[f(\boldsymbol{\theta}^*(\boldsymbol{m}), \boldsymbol{m})\nabla_{\boldsymbol{s}} \ln p(\boldsymbol{m}|\boldsymbol{s}_t)]. \tag{34}$$

And the approximate gradient at iteration $t$ is:

$$\nabla_{\boldsymbol{s}}\Phi_{\text{approx}}(\boldsymbol{s}_t) = f(\boldsymbol{\theta}_{t,k}, \boldsymbol{m}_t)\nabla_{\boldsymbol{s}} \ln p(\boldsymbol{m}_t|\boldsymbol{s}_t). \tag{35}$$

Define the gradient error:

$$e_t = \nabla_{\boldsymbol{s}}\Phi_{approx}(\boldsymbol{s}_t) - \nabla_{\boldsymbol{s}}\Phi(\boldsymbol{s}). \tag{36}$$

Therefore, the outer update becomes:

$$\boldsymbol{s}_{t+1} = \boldsymbol{s}_t - \eta_t(\nabla_{\boldsymbol{s}}\Phi(\boldsymbol{s}_t) + e_t), \tag{37}$$

where $e_t$ is the bias in gradient estimation.

Since $f$ is $K$-Lipschitz in $\boldsymbol{\theta}$ and $\|\nabla_{\boldsymbol{s}} \ln p(\boldsymbol{m}|\boldsymbol{s})\| \leq B$, we have:

$$\mathbb{E}[\|e_t\|] = \mathbb{E}_{\boldsymbol{m}_t}\left[\|f(\boldsymbol{\theta}_{t,k}, \boldsymbol{m}_t)\nabla_{\boldsymbol{s}} \ln p(\boldsymbol{m}_t|\boldsymbol{s}_t) - f(\boldsymbol{\theta}^*(\boldsymbol{m}_t), \boldsymbol{m}_t)\nabla_{\boldsymbol{s}} \ln p(\boldsymbol{m}_t|\boldsymbol{s}_t)\|\right]$$
$$\leq KB\mathbb{E}\left[\|\boldsymbol{\theta}_{t,k} - \boldsymbol{\theta}^*(\boldsymbol{m}_t)\|\right]. \tag{38}$$

For outer loop, we have:

$$\mathbb{E}[\Phi(\boldsymbol{s}_{t+1})] \leq \mathbb{E}[\Phi(\boldsymbol{s}_t)] - \eta_t\langle\nabla_{\boldsymbol{s}}\Phi(\boldsymbol{s}_t), \mathbb{E}[\nabla_{\boldsymbol{s}}\Phi(\boldsymbol{s}_t) + e_t]\rangle + \frac{L_s\eta_t^2}{2}\mathbb{E}[\|\nabla_{\boldsymbol{s}}\Phi(\boldsymbol{s}_t) + e_t\|^2]$$

$$= \mathbb{E}[\Phi(\boldsymbol{s}_t)] - \eta_t\|\nabla_{\boldsymbol{s}}\Phi(\boldsymbol{s}_t)\|^2\left(1 - \frac{L_s\eta_t}{2}\right) - \eta_t\langle\nabla_{\boldsymbol{s}}\Phi(\boldsymbol{s}_t), e_t\rangle\left(1 - L_s\eta_t\right) + \frac{L_s\eta_t^2}{2}\mathbb{E}[\|e_t\|^2]$$

$$\leq \mathbb{E}[\Phi(\boldsymbol{s}_t)] - \eta_t\|\nabla_{\boldsymbol{s}}\Phi(\boldsymbol{s}_t)\|^2\left(1 - \frac{L_s\eta_t}{2}\right) + \eta_t\|\nabla_{\boldsymbol{s}}\Phi(\boldsymbol{s}_t)\| \cdot \|e_t\|\left(1 - L_s\eta_t\right) + \frac{L_s\eta_t^2}{2}\mathbb{E}[\|e_t\|^2]$$

$$\leq \mathbb{E}[\Phi(\boldsymbol{s}_t)] - \eta_t\|\nabla_{\boldsymbol{s}}\Phi(\boldsymbol{s}_t)\|^2\left(1 - \eta_t(\frac{L_s}{2} + KBD) + KBDL_s\eta_t^2(1 - \frac{KBD\eta_t}{2})\right)+$$

$$\eta_t\|\nabla_{\boldsymbol{s}}\Phi(\boldsymbol{s}_t)\|KBD\left(\sqrt{(1 - \mu\alpha)^k}\,\mathbb{E}\left[\|\boldsymbol{\theta}_{t-1} - \boldsymbol{\theta}^*(\boldsymbol{m}_{t-1}))\|\right] + \sqrt{\frac{\alpha\sigma^2}{\mu}}\right)(1 - L_s\eta_t + KBL_s\eta_t^2)+$$

$$\frac{L_s\eta^2}{2}\left(\sqrt{(1 - \mu\alpha)^k}\,\mathbb{E}\left[\|\boldsymbol{\theta}_{t-1} - \boldsymbol{\theta}^*(\boldsymbol{m}_{t-1}))\|\right] + \sqrt{\frac{\alpha\sigma^2}{\mu}}\right)^2, \tag{39}$$

For sufficiently small $\eta_k$, we have $\left(1 - \eta_t(\frac{L_s}{2} + KBD) + KBDL_s\eta_t^2(1 - \frac{KBD\eta_t}{2})\right) > 0$, and the third term is $O(\eta_t^2)$. When $\alpha$ is sufficiently small and $k$ is sufficiently large, the second term becomes sufficiently small. This leads to:

$$\mathbb{E}[\Phi(\boldsymbol{s}_{t+1})] \leq \mathbb{E}[\Phi(\boldsymbol{s}_t)] \tag{40}$$

$\square$

## C. Limitations

Similar to existing approaches, our study did not employ industrial-grade code optimization techniques during the inference evaluation phase. It should be noted that when applying our algorithm to real-world industrial scenarios, the actual acceleration performance may differ from the reported results. Nevertheless, both theoretical analysis and experimental validation demonstrate that our method maintains superior performance compared to existing approaches.

## D. Broader Impacts

While the proposed method demonstrates significant improvements in sampling efficiency and generation quality, it has certain limitations. First, the bilevel optimization framework, though effective, introduces additional computational overhead during the inner-loop fine-tuning, which may limit its scalability for extremely large models or datasets. Second, the performance gains are contingent on the quality of the initial pretrained model and the candidate timestep set, which could vary across different architectures or tasks. Additionally, the policy gradient-based optimization, while avoiding backpropagation, may still require careful hyperparameter tuning to ensure stable convergence. These factors suggest that the method's applicability might be constrained in scenarios with stringent computational budgets or highly specialized domains.

