# OpenReview forum: "Efficient Diffusion Models via Time Step Optimization with Consistent Training and Inference Constraints"
_ICML.cc/2026/Conference — ICML 2026 regular_

### Official Review · Reviewer_Efj3 · 2026-03-03

**Soundness:** 3
**Presentation:** 3
**Significance:** 3
**Originality:** 3
**Overall Recommendation:** 4
**Confidence:** 3

**Summary:**

This paper proposes a novel method for time step selection in diffusion models. The discrete optimization problem over time steps is first reformulated as a continuous optimization problem, enabling gradient-based optimization via policy gradient. A bilevel optimization framework is then introduced to align time step choices between the training and inference stages. Under mild assumptions, theoretical convergence guarantees are established. Numerical experiments demonstrate the effectiveness of the proposed approach.

**Compliance With Llm Reviewing Policy:**

Affirmed.

**Final Justification:**

This paper proposes a novel method for time step selection in diffusion models. The authors have fully addressed my main concerns in the rebuttal. Therefore, I keep my score of 4.

**Key Questions For Authors:**

Please find the questions in ***Strengths and Weakness***.

**Limitations:**

Yes.

**Strengths And Weaknesses:**

Strengths:
- A novel method for optimizing the choice of time steps in training and inference stage.
- Numerical experiments verify the declared contributions.
- The problem addressed——optimizing time step choices to reduce the number of function evaluations (NFE)——is of significant interest to the diffusion modeling community.
- The paper is clearly written and easy to follow.

Weaknesses:
- Numerical experiments:
  - The additional computational overhead introduced by the time step optimization process is not reported, making it difficult to assess the practical trade-offs of the method.
- Presentation:
  - The caption and the content of Table 2 seem to be inconsistent.
  - The naming of the time scheduler differs between Table 2 and Table 4, which may confuse readers.

---

> ### Author Rebuttal · Authors · 2026-03-31
>
> We thank the reviewer for the careful reading and helpful feedback.
>
> > W1: The additional computational overhead introduced by the time step optimization process is not reported, making it difficult to assess the practical trade-offs of the method.
>
> Taking the DDIM sampler as an example, we report the A100 GPU hours for PCS and PCS+FT across four datasets in **Table 3**. It is worth noting that for the version without finetuning, the execution time is primarily dominated by the outer evaluation phase; consequently, the latency is roughly proportional to the number of function evaluations. Furthermore, as we initialize the model with pretrained diffusion parameters and restrict inner training to specific timesteps, the computational overhead of the inner loop is significantly lower than that of full pretraining.
>
> **Table 1: Computational cost (A100 GPU hours) of PCS and PCS+FT across datasets under different NFEs.**
> | Dataset | Methods | NFE=4 | NFE=6 | NFE=8 | NFE=10|
> | :--- | :--- | :---: | :---: | :---: | :---: |
> | **CIFAR-10** | PCS | 0.3 | 0.43 | 0.57 | 0.72 |
> | | PCS+ft | 2.7 | 2.9 | 3.2 | 3.6 |
> | **FFHQ** | PCS | 0.64 | 0.90 | 1.2 | 1.5 |
> | | PCS+ft  | 7.7 | 8.6 | 9.5 | 10.7|
> | **ImageNet** | PCS  | 0.67| 0.95 | 1.3 | 1.6 |
> |  | PCS+ft  | 8.5 | 9.1 | 10.0 | 11.4 |
> | **AFHQv2** | PCS | 0.6 | 0.87 | 1.16 | 1.4 | |
> |  | PCS+ft  | 6.9 | 7.6 | 8.7 | 10.0|
>
> > W2: The caption and the content of Table 2 seem to be inconsistent.
>
> We apologize for the unclear caption and any potential confusion in the initial version. The primary objective of Table 2 is to provide an **empirical motivation** for our work by demonstrating that optimal timestep schedules are highly sampler-dependent, and no single handcrafted rule is universally superior across different solvers.
>
> The correct caption for Table 2 should be:
>
> *Table 2: Comparison of FID scores on CIFAR10 using iPNDM and Uni-PC across different NFEs. We compare commonly used handcrafted timestep schedules under different solvers without additional training. The results show that the effectiveness of handcrafted schedules is highly solver-dependent: no single schedule consistently performs best across different solvers. This observation motivates the need for adaptive timestep optimization.*
>
> As illustrated in the Table 2 (evaluated on CIFAR-10, FID metric):
> - **iPNDM Sampler:** The EDM rule achieves superior generation quality compared to the LogSNR rule (e.g., **5.55** vs. 5.73 at NFE=8).
> - **Uni-PC Sampler:** In sharp contrast, the LogSNR rule significantly outperforms the EDM rule under identical conditions (e.g., **4.41** vs. 9.65 at NFE=8).
>
> We will rewrite the caption for Table 2 and the corresponding discussions in Section 4.1 in the camera-ready version to make the presentation of this research motivation clearer and more intuitive.
>
>
> > W3: The naming of the time scheduler differs between Table 2 and Table 4, which may confuse readers.
>
> We thank the reviewer for pointing this out. We have unified the terminology throughout the paper (e.g., consistently using "LogSNR") to ensure that the naming in Table 2 and Table 4 is consistent and avoids potential confusion. This will be carefully corrected in the camera-ready version.

---

> > ### Author Rebuttal · Reviewer_Efj3 · 2026-04-04
> >
> > I thank the authors for the response. All my concerns are addressed. Therefore, I maintain my positive score.

---

> > > ### Author Response · Authors · 2026-04-04
> > >
> > > We thank the Reviewer for their feedback and for confirming that all concerns have been addressed. We are glad to hear that our responses provided the necessary clarifications, and we appreciate your continued support for our work.

---

### Official Review · Reviewer_5nWL · 2026-03-11

**Soundness:** 3
**Presentation:** 3
**Significance:** 3
**Originality:** 3
**Overall Recommendation:** 4
**Confidence:** 3

**Summary:**

The paper improves the sampling efficiency of diffusion models by learning optimal diffusion time steps. The authors propose probabilistic timestep masking. The time steps are selected through a Bernoulli mask whose parameters are optimized using policy gradient methods. The method introduces a two-step optimization to address the mismatch between training and inference: outer loop for timestep selection, inner loop to fine-tune the diffusion model. Experiments show improvement with extremely small sampling budgets.

**Compliance With Llm Reviewing Policy:**

Affirmed.

**Final Justification:**

My final score remains unchanged as the concerns have been resolved, and I feel assured to keep my positive score.

**Key Questions For Authors:**

What is the computational cost of bilevel optimization compared to standard timestep tuning?

**Limitations:**

yes

**Strengths And Weaknesses:**

Strengths
+ Addresses an important problem: diffusion sampling inefficiency
+ Extensive empirical evaluation across several datasets and samplers
+ Introduces learned timestep selection framework

Weaknesses
- Rather incremental improvements
- Rather limited evaluation

---

> ### Author Rebuttal · Authors · 2026-03-31
>
> We thank the reviewer for the thoughtful evaluation and insightful feedback.
>
> > W1: Rather incremental improvements
>
> - We introduce a framework based on Probabilistic Mask for diffusion sampling. This reparameterization transforms the discrete, non-differentiable timestep selection problem into a continuous and differentiable optimization task, enabling gradient-based search.
> - We identify and tackle the inherent discrepancy between full-trajectory training and sparse-step inference in DPMs. By extending our framework into a Bilevel Optimization paradigm, the inner loop performs lightweight fine-tuning to align the model's capacity with the specific sampling trajectory identified by the outer loop, significantly unlocking performance in few-step scenarios.
> - Unlike traditional coreset selection where bi-level optimization faces an intractable search space due to the sheer volume of samples, our method operates in a much lower-dimensional time-step space (e.g., N=128). This lower dimensionality allows our bi-level framework to converge quickly and reliably, providing a practical and efficient solution specifically designed for diffusion model acceleration.
>
> > W2: Rather limited evaluation.
>
> To address the reviewer's concern, we have significantly expanded our experimental scope:
>
> - **Comprehensive Sampler Evaluation:** In addition to the DDIM and iPNDM samplers reported in Table 3, we have further validated our framework on state-of-the-art solvers, including Uni-PC and DPM-Solver++. The results demonstrate that both PCS and PCS+ft can adaptively identify optimal timestep distributions that cater to the unique numerical properties of different solvers, consistently outperforming conventional handcrafted heuristics.
>
> ### Table 1: Comparison of generation quality (FID ↓) across different samplers and datasets
>
> |Dataset|Sampler|Schedule|NFE=4|NFE=6|NFE=8|NFE=10|
> |:---|:---|:---|:---:|:---:|:---:|:---:|
> |**CIFAR-10**|**Uni-PC**|EDM|50.65|19.56|9.65|6.12|
> |||**PCS**|13.52|5.84|3.49|2.86|
> |||**PCS+ft**|**13.38**|**5.83**|**3.37**|**2.80**|
> ||**DPM-Solver++**|EDM|52.19|12.17|4.67|3.26|
> |||**PCS**|18.79|7.46|3.37|3.11|
> |||**PCS+ft**|**18.31**|**7.33**|**3.32**|**3.02**|
> |**AFHQv2**|**Uni-PC**|EDM|23.74|10.24|7.75|6.26|
> |||**PCS**|13.01|3.77|2.91|2.82|
> |||**PCS+ft**|**12.89**|**3.74**|**2.87**|**2.79**|
> ||**DPM-Solver++**|EDM|19.65|9.64|6.54|4.11|
> |||**PCS**|13.77|5.48|3.61|2.61|
> |||**PCS+ft**|**13.69**|**5.42**|**3.56**|**2.57**|
> |**FFHQ**|**Uni-PC**|EDM|47.78|15.09|11.88|8.79|
> |||**PCS**|20.53|5.98|3.47|3.25|
> |||**PCS+ft**|**20.29**|**5.89**|**3.44**|**3.24**|
> ||**DPM-Solver++**|EDM|38.77|15.17|9.55|7.19|
> |||**PCS**|27.83|7.56|4.47|3.35|
> |||**PCS+ft**|**27.66**|**7.51**|**4.35**|**3.30**|
>
> - **Broader Dataset and Scalability Testing:** We have extended our evaluation of PCS (without finetuning) to datasets with higher resolutions to verify its scalability. Due to the limited time available during the rebuttal period, we have focused on providing the PCS results first. The evaluation for the PCS+ft's version on these additional datasets is currently underway, and we would be pleased to provide these supplementary results in the subsequent discussion phase.
>
> ### Table 2: Comparison of generation quality for PCS on LSUN Bedroom 256×256
> | Solver | Discretization | NFE=4 | NFE=5 | NFE=6 |
> | :--- | :--- | :---: | :---: | :---: |
> | **DPM-Solver++** | LogSNR | 80.44 | 35.81 | 16.95 |
> | | Uniform | 48.82 | 18.64 | 8.50 |
> | | EDM | 324.41 | 294.61 | 268.96 |
> | | PCS | **25.49** | 11.98 | 5.61 |
> | **iPNDM** | LogSNR | 55.77 | 32.51 | 20.26 |
> | | Uniform | 11.93 | 6.38 | 5.08 |
> | | EDM | 312.44 | 284.15 | 252.37 |
> | | PCS | **8.27** | **5.84** | **4.37** |
> | **Uni_PC** | LogSNR | 73.87 | 34.06 | 17.18 |
> | | Uniform | 39.78 | 13.88 | 6.57 |
> | | EDM | 297.83 | 259.90 | 232.79 |
> | | PCS | **18.86** | 9.01 | 4.87 |
>
>
> ### Table 3: Comparison of generation quality for PCS on ImageNet 256×256
> | Solver | Discretization | NFE=4 | NFE=5 | NFE=6 |
> | :--- | :--- | :---: | :---: | :---: |
> | **DPM-Solver++** | LogSNR | 54.61 | 23.24 | 11.52 |
> | | Uniform | 26.07 | 11.91 | 7.51 |
> | | EDM | 244.49 | 233.18 | 221.56 |
> | | PCS | **15.74** | **6.58** | **4.69** |
> | **iPNDM** | LogSNR | 51.35 | 24.93 | 13.94 |
> | | Uniform | 13.86 | 7.80 | 6.03 |
> | | EDM | 237.68 | 223.29 | 210.10 |
> | | PCS | **8.99** | **5.95** | **5.03** |
> | **Uni_PC** | LogSNR | 50.26 | 19.22 | 9.08 |
> | | Uniform | 20.01 | 8.51 | 5.92 |
> | | EDM | 235.31 | 218.15 | 203.26 |
> | | PCS | **9.32** | **4.93** | **4.41** |
>
> Baselines are taken from [r1].
>
> [r1] Learning to Discretize Denoising Diffusion ODEs
>
> > Q1: What is the computational cost of bilevel optimization compared to standard timestep tuning?
>
> Please reference to Weakness 1 of Reviewer **Efj3**.

---

> > ### Author Rebuttal · Reviewer_5nWL · 2026-04-02
> >
> > Thanks for the extensive rebuttal. The concerns have been resolved and I feel assured to keep my positive score.

---

> > > ### Author Response · Authors · 2026-04-03
> > >
> > > We sincerely thank the Reviewer for their positive feedback and for acknowledging that their concerns have been fully resolved. We are pleased to hear that our revisions and clarifications met your expectations. We also appreciate your time and effort in helping us improve the quality of this work.

---

### Official Review · Reviewer_Zeod · 2026-03-13

**Soundness:** 3
**Presentation:** 3
**Significance:** 3
**Originality:** 3
**Overall Recommendation:** 4
**Confidence:** 4

**Summary:**

Summary

This paper studies timestep selection for diffusion model sampling. The authors argue that there is a mismatch between training (which uses thousands of timesteps) and inference (which typically uses far fewer). To address this, they formulate timestep selection as a bilevel optimization problem. The inner loop fine-tunes the model on a subset of timesteps, while the outer loop learns which timesteps to use. To make the discrete selection problem differentiable, they use probabilistic masking and optimize it with policy gradients. Experiments on CIFAR-10, ImageNet, FFHQ, and AFHQv2 with DDIM and iPNDM show consistent improvements over handcrafted schedules and several recent optimization-based methods.

**Compliance With Llm Reviewing Policy:**

Affirmed.

**Key Questions For Authors:**

- How to select the time steps may correspond to the piecewise linear structure of the ODE. It would be helpful to include a toy visualization showing that regions with higher curvature require more evaluation points, while straighter regions require fewer. If the masking mechanism can learn this behavior, it would greatly strengthen the motivation of the paper.
- Can you report the actual computational cost of the bilevel optimization (e.g., GPU hours)?
- How robust is the method to initialization? If you start from a uniform grid, does it still converge to a good schedule?
- Does the learned schedule generalize across different solvers, or do you need to rerun the optimization for each solver?
- Did you observe any instability or variance issue from the policy gradient estimator during training?

**Limitations:**

The main limitation seems to be the computational overhead of the bilevel optimization. The method also assumes access to the original training data and a pretrained model for fine-tuning, which may not always be available. It is also unclear how well the learned schedules transfer across different inference setups (e.g., different solvers or guidance scales).

**Strengths And Weaknesses:**

Strengths

- The motivation is reasonable. The training–inference mismatch is a real issue, and Table 1 gives good evidence that retraining on selected timesteps can help.
- The probabilistic masking + policy gradient formulation is a clever way to avoid differentiating through the entire sampling process.
- Experiments are fairly thorough across multiple datasets and samplers, and improvements over baselines seem consistent, especially at low NFEs.
- Theoretical convergence results are included, which at least give some intuition about the optimization procedure.

Weaknesses

- The bilevel optimization seems expensive. The paper mentions running many inner-loop updates and evaluating large batches during the outer loop, but it’s hard to tell what the actual computational cost is compared to simply using a handcrafted schedule.
- The method appears somewhat sensitive to initialization. EDM initialization performs much better than uniform in Table 6, which suggests the approach still relies on a reasonably good starting schedule.
- The candidate set size has a large effect on performance, but there is no clear guidance on how to choose it.
- The convergence theory relies on assumptions (e.g., strong convexity) that are unlikely to hold for neural network training, so the theoretical guarantees feel somewhat disconnected from practice.

---

> ### Author Rebuttal · Authors · 2026-03-31
>
> We sincerely thank the reviewer for the constructive feedback and valuable comments. We address the weaknesses and questions below.
>
> Due to space limitations, we provide detailed experimental results (Table 1–Table 4) at the following anonymous link:
> https://anonymous.4open.science/r/icml-8C61/README.md
>
> > W1&Q2: Complexity of Bilevel Optimization.
>
> Please see W1 of Reviewer **Efj3**.
>
> > W2&Q3: Sensitivity to initilization.
>
> Thanks. **To clarify, our initialization refers to the discrete distribution of the N candidate timesteps across [0,T]. Our method then selects K steps from these N steps.** While it is reasonable to be influenced by this initialization, our bilevel method is designed to mitigate such sensitivity:
>
> - PCS improves any initialization: Table 1 shows that PCS even starting from a Uniform grid significantly outperforms baseline schedules.
>
> - Bilevel Optimization mitigates sensitivity: Table 1 shows that PCS+FT substantially reduces the initialization gap by adapting model $\theta$ to the selected schedule.
>
> > W3: Candidate set size.
>
> - Feasible Domain vs. Optimization Ceiling: larger $N$ expands the feasible domain $\mathcal{C}$, which raises the theoretical performance ceiling.
> - Saturation phenomenon: Table 2 shows increasing $N$ from 32 to 128 effectively improves FID, further increasing it to 256 yields gains on FID $<0.1$ **on all datasets**.
>
> > W4: Assumption in theory.
>
> Thanks. We can relax the assumption to a general non-convex function, see the theorem below. When the inner loop is fully or partially optimized, our method essentially performs a standard or biased SGD for the outer-loop, which has been extensively studied in previous works, e.g., [r1]. Thus, this theory is not our key contribution and the proof will be included in the revised version due to the space limitation.
>
> Theorem. Assume that $\Phi(\boldsymbol{s})$ is $L$-smooth, and the variance of PGE satisfies
>
> $$ \mathbb{E} || \mathcal{L}_{\mathcal{B}}(\theta^*(m)) \nabla_s \ln p(m|s) - \nabla_s \Phi(s) ||^2 \leq \sigma^2. $$
>
> Let $\eta < 1/L$, and define the gradient mapping at iteration $t$ as:
>
> $$ \mathcal{G}^t = \frac{1}{\eta} ( s^t - \mathcal{P}_C ( s^t - \eta \nabla_s \Phi(s^t) ) ), $$
>
> then the following bound holds as $T \to \infty$:
>
> $$ \frac{1}{T}\sum_{t=1}^T \mathbb{E} |\mathcal{G}^t|^2 \leq \frac{8 - 2L\eta}{2 - L\eta} \sigma^2. $$
>
> [r1] On biased stochastic gradient estimation. JMLR 2022.
>
> > Q1: Correspondence on piecewise linear structure of ODE.
>
> Thanks.  We have the following discussions.
>
> - Curvature and Score Function: The trajectory is governed by the score function $\nabla_{\mathbf{x}} \log p_t(\mathbf{x}_t)$. Near the clean image limit ($t \to 0$), the data distribution becomes highly non-Gaussian and multi-modal, causing the score function's gradient to change abruptly, results in higher curvature $\kappa$, refer to [r2] Fig. 3.
>
> - Why EDM outperforms Uniform: EDM (using a power-law distribution) outperforms Uniform, as it clusters more timesteps in small-noise region ($t \to 0$), i.e., high-curvature phase.
>
> - PCS discovers high-curvature regions: The table 3 shows that our method automatically identifies more error-sensitive steps  concentrated in the later denoising stages, i.e., the high-curvature region, than Uniform and EDM.
>
> [r2] Elucidating the Design Space of Diffusion-Based Generative Models
>
> > Q4: Generalize across solvers.
>
> In principle, the optimal timestep schedule is solver-dependent because different  solvers  exhibit distinct local truncation error characteristics.  Our PCS is designed to capture these specific error patterns to minimize the global discretization error.
>
> Table 4 shows reasonable cross-solver transfer at NFE=10, while NFE=4 is more sensitive. Since PCS is efficient (~0.5h), we recommend re-optimization per solver.
>
> > Q5: Variance of policy gradient (PGE)?
>
> We did not observe significant high-variance issues during training due to the following reasons:
>
> - Unlike general coreset selection tasks that often involve optimizing over millions of samples, our candidate set size is relatively small ($N \approx 128$). This low-dimensional space significantly reduces the variance of PGE.
>
> -  We empirically found that standard variance reduction techniques were unnecessary.
>
> > Limitation:
>
> We thank the reviewer for pointing out these limitations, which we will thoroughly discuss in the camera-ready version. Regarding the access to model parameters, we would like to clarify that if our method is restricted to the outer loop for timestep selection, it does not require access to the model's internal parameters or gradients. In this mode, the model can be treated as a black box, and the optimization only relies on the generation outputs to guide the policy gradient.

---

> > ### Author Rebuttal · Reviewer_Zeod · 2026-04-03
> >
> > Thank you for the detailed and helpful rebuttal. I appreciate that.
> >
> > These responses address most of my concerns. In particular, it is helpful to see that the method is reasonably robust to initialization and that the optimization overhead is relatively small in practice.
> >
> > That said, my overall assessment remains unchanged. While the method is technically sound and well-motivated, its practical impact appears somewhat limited. In particular, recent one-step or few-step generation methods already achieve strong performance, which may reduce the need for carefully optimized timestep schedules in many settings.
> >
> > Overall, I believe the concerns have been largely addressed, but the scope of impact remains somewhat narrow, so I keep my original evaluation.

---

> > > ### Author Response · Authors · 2026-04-04
> > >
> > > We sincerely thank the reviewer for the positive update and for recognizing both the motivation and the technical soundness of our work. We are especially encouraged that the rebuttal helped clarify two central concerns, namely the robustness of our method to initialization and the relatively small optimization overhead in practice. We also appreciate the reviewer’s acknowledgment that these responses addressed most of the original questions.
> > >
> > > We agree that recent one-step and few-step generators are an exciting direction for fast generation [r1, r2]. However, in practical settings such as controllable generation and editing, recent works still report clear limitations of one-step models, as well as continued sensitivity to the underlying noise or sampling schedule [r8, r9, r10]. Therefore, we believe our study remains valuable in practical applications for the following three main reasons.
> > >
> > > First, these approaches typically rely on additional distillation or specialized retraining, whereas our method improves already pre-trained diffusion models without retraining and is therefore directly applicable in a broader plug-and-play setting [r1, r2, r8].
> > >
> > > Second, recent literature continues to show that schedule design remains impactful, especially in the few-step regime: optimized schedules consistently outperform hand-crafted heuristics across continuous diffusion, image and video generation, as well as discrete diffusion settings [r3, r4, r5, r6, r7, r9].
> > >
> > > Third, this direction is orthogonal rather than competing with one-step generation, since stronger few-step samplers and better teacher trajectories are exactly what modern distillation methods compress [r1, r2]. In this sense, improving multistep sampling and schedule design can also benefit future one-step or few-step distillation pipelines [r1, r2, r8].
> > >
> > > [r1] T. Salimans and J. Ho. Progressive Distillation for Fast Sampling of Diffusion Models. In International Conference on Learning Representations (ICLR), 2022.
> > >
> > > [r2] Y. Song, P. Dhariwal, M. Chen, and I. Sutskever. Consistency Models. In Proceedings of the 40th International Conference on Machine Learning (ICML), 2023.
> > >
> > > [r3] T. Karras, M. Aittala, T. Aila, and S. Laine. Elucidating the Design Space of Diffusion-Based Generative Models. In Advances in Neural Information Processing Systems 35 (NeurIPS 2022), 2022.
> > >
> > > [r4] C. Lu, Y. Zhou, F. Bao, J. Chen, C. Li, and J. Zhu. DPM-Solver++: Fast Solver for Guided Sampling of Diffusion Probabilistic Models. Machine Intelligence Research, 2025.
> > >
> > > [r5] A. Sabour, S. Fidler, and K. Kreis. Align Your Steps: Optimizing Sampling Schedules in Diffusion Models. In Proceedings of the 41st International Conference on Machine Learning (ICML), 2024.
> > >
> > > [r6] S. Xue, Z. Liu, F. Chen, S. Zhang, T. Hu, E. Xie, and Z. Li. Accelerating Diffusion Sampling with Optimized Time Steps. In Proceedings of the IEEE/CVF Conference on Computer Vision and Pattern Recognition (CVPR), 2024.
> > >
> > > [r7] Y.-H. Park, C.-H. Lai, S. Hayakawa, Y. Takida, and Y. Mitsufuji. Jump Your Steps: Optimizing Sampling Schedule of Discrete Diffusion Models. In International Conference on Learning Representations (ICLR), 2025.
> > >
> > > [r8] Y. Luo, T. Hu, Y. Song, J. Sun, Z. Li, and J. Tang. Adding Additional Control to One-Step Diffusion with Joint Distribution Matching. In Proceedings of the IEEE/CVF International Conference on Computer Vision (ICCV), 2025.
> > >
> > > [r9] H. Lin, Y. Chen, J. Wang, W. An, M. Wang, F. Tian, Y. Liu, G. Dai, J. Wang, and Q. Wang. Schedule Your Edit: A Simple yet Effective Diffusion Noise Schedule for Image Editing. In Advances in Neural Information Processing Systems (NeurIPS 2024), 2024.
> > >
> > > [r10] A. Ravishankar, S. Liu, M. Wang, T. Zhou, J. Zhou, A. Sharma, Z. Hu, L. Das, A. Sobirov, F. Siddique, F. Yu, S. Baek, Y. Luo, and M. Wang. Fair Benchmarking of Emerging One-Step Generative Models Against Multistep Diffusion and Flow Models. arXiv preprint arXiv:2603.14186, 2026.

---

### Decision · Program_Chairs · 2026-04-30

**Decision:**

Accept (regular)

**Comment:**

This paper proposes an optimization scheme for the timestep selection during diffusion inference. While the problem is motivated by few step and low NFE approaches, there is currently no discussion of consistency maps or self-distillation, which strikes me as an oversight. Nevertheless, the theoretical foundations of the approach are clear and well motivated and the empirical results are strong. The reviews all emphasize the results and the approach are sound and well explained.